# Exploiting hosts and vectors: viral strategies for facilitating transmission

Xi Yu[1,2,3,4,8], Yibin Zhu[1,3,8], Gang Yin[5], Yibaina Wang[6], Xiaolu Shi[3] & Gong Cheng [ID][1,2,3,7 ✉]

## Abstract

**Viruses have developed various strategies to ensure their survival and transmission. One intriguing strategy involves manipulating the behavior of infected arthropod vectors and hosts. Through intricate interactions, viruses can modify vector behavior, aiding in crossing barriers and improving transmission to new hosts. This manipulation may include altering vector feeding preferences, thus promoting virus transmission to susceptible individuals. In addition, viruses employ diverse dissemination methods, including cell-to-cell and intercellular transmission via extracellular vesicles. These strategies allow viruses to establish themselves in favorable environments, optimize replication, and increase the likelihood of spreading to other individuals. Understanding these complex viral strategies offers valuable insights into their biology, transmission dynamics, and potential interventions for controlling infections. Unraveling interactions between viruses, hosts, and vectors enables the development of targeted approaches to effectively mitigate viral diseases and prevent transmission.**

**Keywords** Viral Transmission Strategies; Viral Transmission Dynamics; Vector Behavior; Host-virus Interactions
**Subject Category** Microbiology, Virology & Host Pathogen Interaction

## Introduction

Viruses are obligate intracellular parasites that rely entirely on their eukaryotic or prokaryotic hosts throughout their replication cycle. These hosts play a vital role in providing the essential resources and infrastructure required by the virus to replicate, package itself, and spread within the host organism. However, after completing replication and exhausting the host's resources, viruses face the challenge of crossing physical and spatial barriers in order to infect new hosts. This critical process, known as transmission, involves the implementation of specific strategies, which can be broadly categorized into two main forms: vertical and horizontal transmission. Vertical transmission refers to the direct transmission of a pathogen from an infected parent to the subsequent generation (Dahiya et al, 2022; Recaioglu and Kolk, 2023). Conversely, horizontal transmission encompasses the transfer of the virus between individual hosts within the same species or conceivably among species that lack a familial relationship (Sabeena and Ravishankar, 2022). In its simplest manifestation, viruses benefit from the unrestricted movement through the surrounding medium or extracellular matrix via Brownian motion, ultimately reaching new host cells. This vector-less and support-less transmission is observed primarily in viruses infecting microorganisms like virophages and bacteriophages. Another mode of transmission is achieved through direct or indirect contact, including instances where viruses are transmitted through open wounds. Airborne transmission occurs when viruses utilize inert carriers to reach and infect new hosts. A case in point involves human rhinoviruses and other similar viruses that can be detected within aerosols generated through sneezing, subsequently facilitating their inhalation by unsuspecting individuals. In addition, certain viruses hitchhike on dust particles and other solid materials to facilitate their dispersal. All alternative modes of horizontal transmission depend on vectors, which are living organisms that facilitate the transfer of viruses between two hosts by bridging the spatial divide. Vectors offer advantages due to their active and purposeful seeking of new hosts, unlike passive distribution mechanisms such as wind or water. Moreover, numerous insect vectors are parasitic and rely on sap or blood feeding, allowing them to directly introduce viruses into the cells and vessels of new hosts.

The transmission process plays a pivotal role in the dissemination of viruses and their infection cycles within the environment. Hence, it is not surprising that viruses have evolved intricate tactics to ensure the efficiency of this process. These tactics involve two fundamental components: firstly, the careful selection of a suitable means of transportation, often relying on vectors as carriers to facilitate the transfer of pathogens between hosts and bridge the gap between them; and secondly, the intentional manipulation exerted by viruses on both the vectors and the hosts, with the aim of optimizing the acquisition and transmission of the pathogen to a new host. Although the deliberate manipulation of hosts and vectors has been extensively studied in the context of eukaryotic parasites (Marzal et al, 2022; Picciotti et al, 2023; Sanford and Shutler, 2022), it is crucial to recognize that viruses should not be overlooked within this framework. This review aims to delve into

[1]New Cornerstone Science Laboratory, Tsinghua-Peking Center for Life Sciences, School of Basic Medical Sciences, Tsinghua University, Beijing 100084, China. [2]Institute of Infectious Diseases, Shenzhen Bay Laboratory, Shenzhen, Guangdong 518000, China. [3]Institute of Pathogenic Organisms, Shenzhen Center for Disease Control and Prevention, Shenzhen, Guangdong 518055, China. [4]School of Life Sciences, Tsinghua University, Beijing 100084, China. [5]Department of Parasitology, School of Basic Medical Sciences, Central South University, Changsha, Hunan 410013, China. [6]China National Center for Food Safety Risk Assessment, Beijing 100022, China. [7]Southwest United Graduate School, Kunming 650092, China. [8]These authors contributed equally: Xi Yu, Yibin Zhu. ✉E-mail: gongcheng@mail.tsinghua.edu.cn

the remarkable strategies employed by viruses to ensure their survival and successful transmission.

# Viral implications on host and vector behaviors: influences on transmission dynamics

## Potential host behavioral alterations triggered by viral infections

Diverse infectious agents possess the capacity to impact the central nervous system of their host, resulting in modifications to the host's behavior. The mechanisms underlying parasite-induced alterations in host behavior are proposed to involve both direct and indirect means (Gowda et al, 2023). However, a comprehensive understanding of the intricate molecular mechanisms and adaptive significance of the behavioral changes triggered by viruses remain limited, and empirical evidence to substantiate these hypotheses is lacking. Rabies virus infection, in particular, is recognized for its pronounced behavioral changes and presents a significant public health concern due to the absence of satisfactory treatment options (Fisher et al, 2018). The manipulation hypothesis posits that infectious agents induce changes in host behavior to enhance their transmission to other susceptible hosts (Lian et al, 2022). Nevertheless, discerning the adaptive capacity of pathogens to manipulate the host from general sickness behaviors presents challenges. Several hypotheses have been postulated to elucidate how parasitic infections specifically alter host behavior, encompassing potential structural damage to critical regions of the central nervous system and immune-related pathological mechanisms (Oswald et al, 2017). In the case of rabies, a disease characterized by severe behavioral changes and neurological disorders, the pathological alterations observed in the brain are typically minimal, rendering a comprehensive understanding of the underlying causes behind these profound behavioral modifications elusive.

The nomenclature of the rabies virus (RV) is thought to be derived from the ancient Indian term "rabh," signifying "to induce acts of violence" (Wang et al, 2018). Subsequent to its infiltration of peripheral regions, the virus proceeds in a retrograde trajectory within the nervous system until it ultimately arrives at the cerebral cortex. In both human beings and animals, the principal cerebral regions that are targeted encompass the limbic system and the hippocampus, characterized by a notably elevated density of RV receptors, including nicotinic acetylcholine receptors (nAChR) (Stein et al, 2010) and metabotropic glutamate receptor subtype 2 (GRM2) (Wang et al, 2018). In the case of animals, this RV infection is succeeded by a substantial release of the virus, presumably originating from neuronal sources, into the saliva, accompanied by the onset of belligerent conduct, commonly denoted as "furious rabies." This aggressive conduct can potentially facilitate the transmission of the infection to new hosts. The nicotinic acetylcholine receptors (nAChR), such as α4β2 nAChR, are involved in locomotor activity and serotonin neurotransmitter function (Lewis et al, 2015), both of which play a role in rabies pathogenesis and behavioral changes (Jackson 2016). The nAChR is a receptor for the neurotoxin-like region of the rabies virus glycoprotein, which inhibits the activity of nAChRs in the CNS, resulting in behavioral alterations in infected animals (Lian et al,

2022; Kumar et al, 2007). This mechanism might enable rabies and potentially other pathogens to influence the behavior of their hosts by targeting nAChR and maybe other to be identified neurotransmitter receptors.

In addition to the rabies virus discussed above, another member of the lyssavirus family, referred to as Borna disease virus (BDV), infects various vertebrate species (Readhead et al, 2018). BDV infection in several non-human species results in enduring, persistent infection. This virus, similar to the rabies virus, exhibits a predilection for the hippocampus. In controlled experiments involving animals, it has been associated with manifestations of anxiety and aggression, even in the absence of fever (Carbone, 2001) (Fig. 1). In addition, late-stage infection has been correlated with impairment to the hippocampal dentate gyrus (Ludlow et al, 2016).

A significant portion of the human population carries various types of herpes viruses (HSV). The distinctive propensity of HSV, particularly HSV-1 and HSV-2, to target the brain has been acknowledged for nearly a century (Kotsiri et al, 2023) (Fig. 1). These viruses can endure within sensory ganglia throughout an individual's lifetime and latent virus is frequently detected in the amygdala, hippocampus, and olfactory system. The hippocampus, having the highest concentration of HSV-1 receptors (Lathe and Haas, 2017), has emerged as a pivotal site for HSV replication during encephalitis. HSV-1 infection has been correlated with localized cell death (apoptosis) in the hippocampus and there have been several instances of Klüver–Bucy syndrome or autism reported following encephalitis caused by HSV, suggesting potential virus-induced harm to the limbic brain (Zappulo et al, 2018). Behavioral alterations during severe infection have been deliberated upon as factors that could conceivably facilitate the transmission of the virus. During the latency phase of HSV-1 within the central nervous system (CNS), there is an associated sustained increase in the levels of cytokines (Baker et al, 1999; Sehl-Ewert et al, 2022). This persistent production of specific cytokines during latency has the potential to have adverse effects on both endocrine function and immune responses. This effect may occur by targeting receptors expressed within the brain and immune cells, although the specific receptors remain undiscovered at present and await further research. There has long been speculation about the potential connection between chronic infections and the occurrence of both depression and anxiety (Prusty et al, 2018; Toro et al, 2019). Human herpesvirus infection has been linked to major depressive disorders. Furthermore, subclinical infections, such as those involving HSV, lead to a sustained elevation in the levels of circulating cytokines, which predominantly act upon regions of the limbic brain, notably the hippocampus (Lathe and Lathe, 2020). The link between cytokines and behavioral changes is supported by a body of evidence that clinical administration of interferons and interleukins, such as TNF-α, IFN-α, IFN-β, IL-1α, and IL-2, induces a state resembling malaise and sickness behavior, which mirrors the symptoms of anxiety and depression (Exton et al, 2002; Raison et al, 2005). In fact, subclinical infections of various types, not necessarily confined to the brain, that trigger systemic inflammation, may potentially contribute to the development of depressive and anxiety disorders (Bullmore, 2018).

Traditionally, the interaction between viruses and receptors, as well as their impact on the host, has been primarily associated with viral cell entry and immune recognition. However, the recent

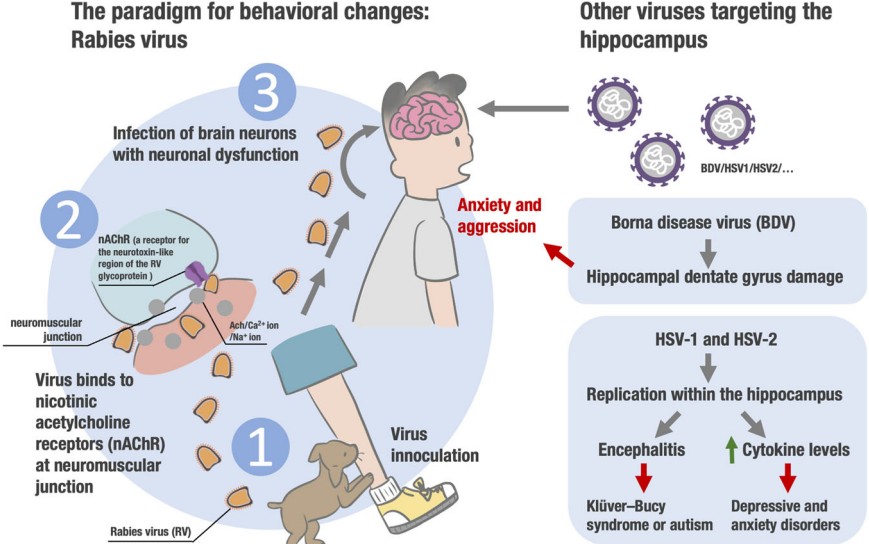

**Figure 1. Host behavioral alterations induced by viral infections.**

Infections can affect the central nervous system, leading to potential behavioral changes. Rabies virus, for instance, causes pronounced aggression, potentially aiding its transmission. Nicotinic acetylcholine receptors play a role in rabies-induced behavioral changes. Similarly, Borna disease virus induces anxiety and aggression by targeting the hippocampus. Herpes simplex viruses also target the hippocampus, potentially leading to behavioral alterations and contributing to depressive and anxiety disorders.

findings discussed above suggest a hypothesis that viruses might be able to influence the central nervous system (CNS) by potentially targeting various neurotransmitter receptors in infected hosts. While evidence exists for the involvement of nAChRs in the context of rabies, further investigation is required to determine if other neurotransmitter receptors bind viral proteins and are modulated by them. This discovery broadens our understanding of virus-host interactions, encompassing not only viral cell entry and immune recognition but also targeted manipulation of the CNS through neurotransmitter receptors. These insights may pave the way for novel therapeutic strategies that target virus-derived peptides in the CNS.

## Vector behavioral alterations induced by viral infections

Arthropods such as mosquitoes, ticks, tsetse flies, aphids, whiteflies, and thrips are common vectors for the transmission of viruses, both in vertebrates and plants, through their feeding on blood or phloem sap. These viruses are collectively referred to as arboviruses or arthropod-borne viruses (Wu et al, 2019). The interaction between viruses and vectors can take place through either circulative or noncirculative transmission. Circulative transmission involves the acquisition of the virus by the vector during feeding, followed by its transport through the haemolymph and eventual transmission to a new host through the salivary glands during feeding (Dader et al, 2017). On the other hand, noncirculative transmission occurs when the virus binds to the mouthparts of the vector without replication or internalization. Noncirculative transmission is observed in various plant and vertebrate viruses, including multiple plant viruses from the *Cucumoviridae* and *Potyviridae* families, as well as vertebrate viruses from the *Caliciviridae* and *Poxviridae* families (Drucker and Then, 2015). In vertebrate viruses, noncirculative transmission is also known as mechanical transmission, as these

viruses can be transmitted through direct physical contact between the vector and the host or between hosts. The mechanisms underlying circulative and noncirculative transmission of viruses differ in their reliance on virus-vector interactions. While circulative transmission entails the passage through intestinal barriers and invasion of salivary glands, the mechanisms underlying noncirculative transmission are less comprehensively understood (Dader et al, 2017). In noncirculative transmission, the adherence of virus particles to the mouthparts of the vector is commonly perceived as a result of mechanical contamination. Nonetheless, noncirculative transmission also exhibits a degree of specificity towards vectors, indicating that it is not a random phenomenon. Moreover, the presence of viral proteins specifically designed to bind to vectors implies that noncirculative transmission is more intricate than previously acknowledged (Blanc et al, 2014).

In addition to the specific molecular interactions that facilitate the attachment and transportation of viruses by insect vectors, researchers have observed that viruses possess the remarkable ability to manipulate these vectors in order to enhance their own transmission. This phenomenon is particularly evident in the case of plant viruses, which can influence the interactions between vectors and hosts through both direct effects, stemming from the presence of virus particles within the vector, and indirect effects, facilitated by viral interference with the host plant (Fig. 2A). Insects, such as aphids, engage in a sophisticated host plant selection process that incorporates various stimuli and responses (Fereres and Moreno, 2009). The transmission process comprises four sequential stages, namely attraction to the host plant, evaluation of the leaf surface, penetration of tissues, and consumption of phloem sap. The physiological changes occurring in the host plant as a consequence of viral infection enable plant viruses to modify specific characteristics of the host plant traits, thereby impacting each stage of this transmission process. Plant

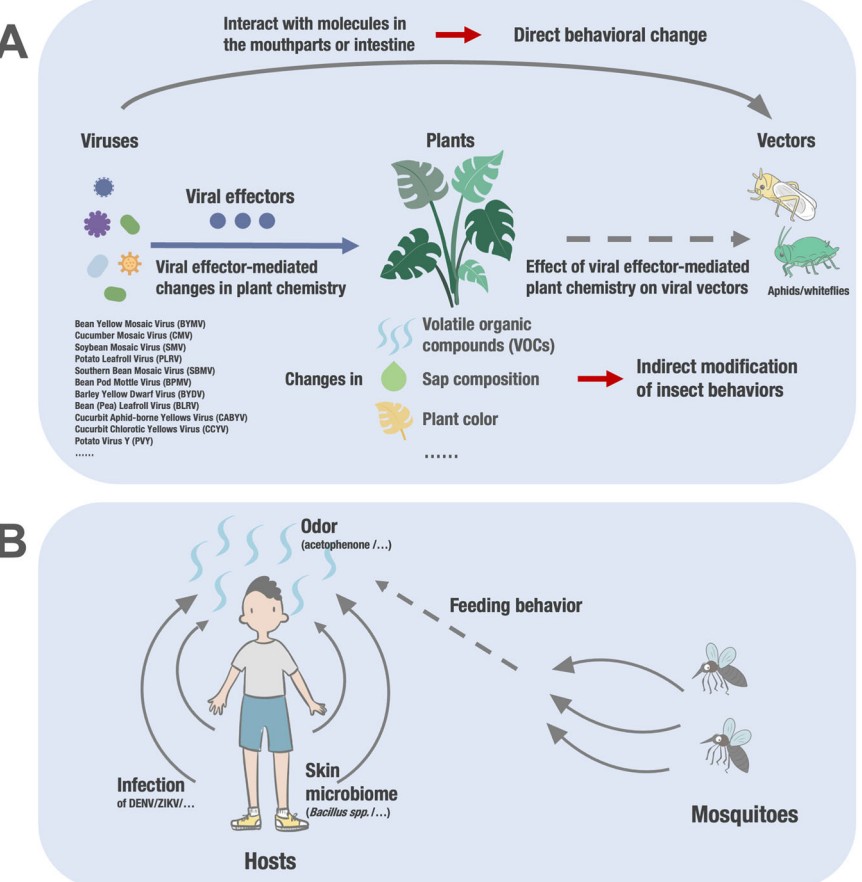

**Figure 2. Vector behavioral alterations induced by viral infections.**

Virus infection in plants (**A**) and vertebrate hosts (**B**), as well as in the insect vector itself (**A**), induces changes in insect feeding preferences, influencing virus transmission. (**A**) This figure illustrates how viruses manipulate insect vectors to enhance transmission. Plant viruses modify host plants, altering volatile organic compounds (VOCs) and leaf color to attract or repel vectors. For instance, BYMV makes plants more attractive to *Acyrthosiphon pisum*, while CMV initially attracts *Aphis gossypii* before they move to healthy plants. Noncirculative viruses cause rapid plant degradation, promoting vector dispersal and rapid virus acquisition, while circulative viruses slow plant decline, encouraging prolonged feeding, as seen with *Myzus persicae* preferring PLRV-infected leaves. Viruses can also directly alter vector behavior. For example, BYDV-infected aphids prefer non-infected plants, and CCYV-infected *Bemisia tabaci* whiteflies show distinct behaviors compared to non-infected ones. These behavioral changes often result from virus interactions with the vector's gut, salivary glands, or nervous system. Vector fitness is impacted differently by virus type. Noncirculative viruses generally reduce plant quality, negatively affecting vector fitness, though benefits are sometimes observed. Circulative viruses typically enhance vector fitness by improving host plant quality, as demonstrated by increased growth rates of *Myzus persicae* on PLRV-infected plants. These interactions between viruses, vectors, and host plants are crucial for efficient viral transmission. (**B**) This figure illustrates how arboviruses manipulate their transmission by altering host attractiveness to vectors. Recent studies suggest that infected hosts can enhance their appeal to vectors through changes in volatile compound release. Investigating flaviviruses such as dengue and Zika viruses, it was found that they manipulate the skin microbiome to attract *Aedes* mosquitoes, with acetophenone emitted from infected hosts acting as a potent attractant. *Bacillus spp.*, known acetophenone producers, proliferate within the skin microbiome during infection, facilitated by the downregulation of the *Retnla* gene encoding the antimicrobial protein RELMα. This selective manipulation of the skin microbiome enables viruses to maximize their transmission without compromising host health significantly. The involvement of commensal skin microbes presents new avenues for intervention in disease transmission, suggesting promising strategies like modifying the microbiome composition to reduce host attractiveness to vectors.

viruses have been found to exert influence over the composition of volatiles emitted by infected hosts (Mauck et al, 2010, 2014; Medina-Ortega et al, 2009; Oluwafemi et al, 2011; Rajabaskar et al, 2014), the composition of sap or leave color, thereby affecting the perception of vectors and manipulating their attraction or repulsion to feed on the infected host (Fereres and Moreno, 2009). Remarkably, certain plant viruses, especially those that circulate within the vector's body or infect the vector itself, have been documented to possess the ability to directly modify vector behavior, further enhancing their chances of successful transmission (Stafford et al, 2011). Ingwell et al proposed the Vector

Manipulation Hypothesis (VMH) as a conceptual framework to explain this phenomenon (Ingwell et al, 2012).

Virus infection can alter plant color and volatile organic compounds (VOCs), influencing the attractiveness of infected plants to insect vectors (Fereres and Moreno, 2009). Once insects land on infected plants, they evaluate their suitability as hosts and decide whether to stay or leave. Both noncirculative and circulative viruses are expected to increase vector attraction to infected plants for efficient virus acquisition. However, different feeding arrestment periods and behaviors are likely required for optimal pathogen spread between noncirculative and circulative viruses

(Mauck et al, 2012). Noncirculative viruses cause rapid plant degradation and low nutrition, discouraging prolonged settlement and promoting vector dispersal (Hodge and Powell, 2008). This facilitates rapid virus acquisition but limits transmission to short periods after probing infected plants. In contrast, circulative viruses benefit from slower decline in plant quality to encourage sustained vector feeding, as they require longer acquisition periods (Mauck et al, 2012). Numerous studies support or challenge models of noncirculative virus-induced changes in plant manipulation by vectors. For example, *Acyrthosiphon pisum* displayed a preference for leaf disks with Bean yellow mosaic virus (BYMV) infection in free-choice assays (Hodge and Powell, 2008). *Aphis gossypii* initially preferred CMV-infected plants but later shifted to healthy plants, suggesting initial acquisition from infected plants and subsequent migration to inoculate healthy plants (Carmo-Sousa et al, 2014). *Aphis glycines* were attracted to Soybean mosaic virus (SMV)-infected plants, exhibiting sustained feeding (Penaflor et al, 2016), while *Myzus persicae* and *Rhopalosiphum maidis* did not find SMV-infected soybean plants attractive (Fereres et al, 1999). However, these aphids stayed for a shorter duration before leaving, increasing the probability of probing and infecting another healthy host before losing virus transmissibility. Studies on circulative viruses have primarily focused on *Luteoviridae* family viruses. *Myzus persicae* preferred leaves infected with Potato leafroll virus (PLRV) over noncirculative viruses, as confirmed by olfactometer tests and emigration bioassays (Alvarez et al, 2007; Eigenbrode et al, 2002; Rajabaskar et al, 2013; Werner et al, 2009). Similar attraction mediated by VOCs was observed in other *Luteoviridae*-aphid interactions, indicating a characteristic feature of circulative viruses, particularly those from the *Luteoviridae* family.

Regarding vector fitness, noncirculative viruses generally reduce host plant quality, negatively impacting aphid fitness (Mauck et al, 2014). However, beneficial effects of noncirculative virus infection have also been reported, such as increased maturity and weight of *M. persicae* on infected Chinese cabbage (Hodgson, 1981). Observations have demonstrated that there are variations in fitness during different stages of infection, where higher fitness is observed in plants recently infected compared to plants that have been infected for a longer period (Casteel et al, 2015; Casteel et al, 2014). This phenomenon aligns with the rejection of older infected plants, indicating a preference for transmission. Circulative viruses enhance aphid fitness by improving the quality of the plant host, promoting prolonged feeding and optimizing virus acquisition. *M. persicae* showed increased growth rates on PLRV-infected potato plants (Castle and Berger, 1993). Other viruses have been observed to exhibit similar positive effects on vector fitness. However, it is important to note that contrasting results have also been documented in some cases (Fiebig et al, 2004). Circulative viruses generally increase vector fitness, particularly for aphid vectors. There is a scarcity of data regarding the impact of virus infection on vector fitness in other hemipteran vectors. Nonetheless, the available reports are consistent with the findings observed in aphid vectors, which can be attributed to the similarities in feeding behaviors among phytophagous hemipterans. Improved vector performance on infected plants may not be specific to a particular system, as demonstrated by the Mexican bean beetle feeding on bean plants infected with southern bean mosaic virus (SBMV) or bean pod mottle virus (BPMV) (Musser et al, 2003). Studies using the electrical penetration graph (EPG) technique have identified modified probing and feeding behaviors in aphids associated with enhanced transmission of both circulative and noncirculative viruses (Alvarez et al, 2007; Carmo-Sousa et al, 2014; Fereres and Moreno, 2009).

In addition to the indirect modification of insect behaviors and performance through plant-mediated effects, scientific investigations have revealed that viruses present in insect vectors can induce behavioral changes that facilitate the transmission process. For instance, aphids infected with Barley yellow dwarf virus (BYDV) show a preference for feeding on non-infected wheat plants, while non-infected aphids prefer BYDV-infected plants (Ingwell et al, 2012; Medina-Ortega et al, 2009). Similar observations have been made in other aphid-related pathosystems, such as *M. persicae*/pea leafroll virus and *A. gossypii*/cucumber aphid-borne yellows virus (Carmo-Sousa et al, 2016; Rajabaskar et al, 2014). These altered preferences, depending on the viral status of the aphids, can promote the spread of the pathogen by facilitating virus acquisition and inoculation. Volatile compounds may play a role in influencing these preferences (Medina-Ortega et al, 2009; Ngumbi et al, 2007). Studies investigating the direct alteration of behavior in other virus-transmitting vector species have yielded similar outcomes. These instances underscore the capability of circulative viruses to modify the behavior of vectors, possibly through interference or interaction with diverse components of the insects during their passage through the vector. The precise mechanisms underlying these relationships remain undisclosed; nevertheless, potential targets for viral interaction may encompass the gut, salivary glands, or nervous system of the vectors. While the impact of non-circulative viruses on insect behaviors is not well understood, it is conceivable that these viruses also interact with molecules in the mouthparts or intestine, thus affecting feeding behavior. Supporting evidence for this hypothesis comes from studies reporting distinct feeding behavior in aphids transmitting potato virus Y (PVY) compared to non-transmitting aphids from the same infected plants (Collar et al, 1997). Likewise, *Bemisia tabaci* whiteflies that were infected with the non-circulative cucurbit chlorotic yellows virus displayed distinct behaviors in comparison to whiteflies that were free of the virus (Lu et al, 2017). In essence, virus infection in plants, regardless of whether it is circulative or noncirculative, induces specific modifications in insect feeding behaviors that subsequently influence the transmission of the virus.

Similarly, investigations have been conducted on the modification of vector behavior by arboviruses that infect vertebrates, and in certain instances, these changes have been linked to alterations in the composition of saliva proteins due to viral infection of the vector (Chisenhall et al, 2014; Tchankouo-Nguetcheu et al, 2012). Conversely, less attention has been given to the virus-induced alteration of attractiveness in infected vertebrate hosts. A number of studies have presented compelling evidence that hosts infected with pathogens can potentially enhance their appeal to vectors of significance through alterations in host cues subsequent to infection (Hughes et al, 2014; Zhang et al, 2022) (Fig. 2B). Nevertheless, further work is required to ascertain whether arboviruses employ a comparable strategy to manipulate interactions between hosts and vectors. Viruses face significant evolutionary pressure to enhance their ability to attract vectors. To enhance vector attraction, one approach involves stimulating infected hosts to release volatile compounds (Pierson and Diamond, 2020). Plasmodium falciparum, the eukaryotic pathogen causing malaria, has been observed

to induce the production of aldehydes in the red blood cells of infected individuals (Robinson et al, 2018). In a recent study, our lab investigated how dengue and Zika viruses enhance their transmission potential by influencing mosquito feeding behavior on infected hosts (Zhang et al, 2022). They discovered that the virus can manipulate the composition of the skin microbiome to attract vectors. *Aedes* mosquitoes did not show a heightened preference for infected hosts in the early stage of flavivirus infection. However, between days 4 and 6, mosquitoes exhibited significantly greater attraction to mice infected with dengue and Zika viruses. Several molecules, including acetophenone, emitted from the skin of infected animals acted as attractants for mosquitoes. While acetophenone can be produced by commensal bacteria residing on the skin and in the intestinal tract (The Human Microbiome Project C (2012)), the skin microbiome, rather than the gut microbiome, played a key role in this attraction. *Bacillus spp.*, known for their production of acetophenone, showed increased abundance within the skin microbiome during infection. This coincided with changes in gene expression in the skin epidermis, particularly a downregulation of the *Retnla* gene encoding the antimicrobial protein resistin-like molecule-α (RELMα), which targets *Bacillus spp*. Through this mechanism, the virus selectively manipulated the skin microbiome without causing significant disruptions in other areas that could compromise the host's health. This sophisticated mechanism enables mosquito-borne flaviviruses to maximize their own transmission. The involvement of commensal skin microbes in the infection process presents new avenues for intervention in disease transmission. Promising strategies, such as modifying the composition of the microbiome to reduce host attractiveness to vectors, merit further exploration.

## Diverse viral transmission strategies

### Cell-to-cell transmission

Enveloped viruses are characterized by the presence of lipid bilayer and depend on host cells for their replication. Within host cells, they undergo genetic replication, assembly of viral progenies, and subsequent release to infect adjacent cells, thereby facilitating the dissemination of the viral infection. It has been established that numerous viruses have the ability to efficiently disseminate through direct cell-to-cell contact (Sanjuan and Thoulouze, 2019). Notably, certain viruses such as herpesviruses, rhabdoviruses, and the measles virus disseminate along neuronal networks, implying transmission through neurological synapses (Lawrence et al, 2000). The ability of African swine fever viruses and vaccinia viruses to initiate the formation of actin tails in infected cells indicates a comparable dissemination mechanism to certain bacteria, such as *Listeria* (Jouvenet et al, 2006). Initially, evidence for cell-to-cell contact in viral spread was indirect, as the limited infectivity of cell-free viruses did not account for their rapid propagation in cell culture (Bangham, 2003). However, electron micrographs and time-lapse movies have provided compelling visual evidence of cell-to-cell transmission (Hubner et al, 2009; Jin et al, 2009; Sherer et al, 2007).

Another notable finding in this field pertains to the intriguing observation that the addition of a small number of dendritic cells (DCs) to a T cell culture considerably enhances the infection of T cells by HIV (Cameron et al, 1992). Subsequent investigations revealed that DCs capture and present HIV to T cells in a manner similar to antigen presentation facilitated by antigen-presenting cells (APCs) (Geijtenbeek et al, 2000; McDonald et al, 2003). This discovery has led to the proposition of a model suggesting that immunotropic viruses, including HIV, exploit immunological synapses to achieve efficient cell-to-cell transmission.

Viruses employ two propagation mechanisms: cell-free dissemination and cell-associated transmission through direct cell-to-cell contact (Fig. 3A). Both methods are effective for viral spread but require specific conditions for success. Cell-free propagation relies on the release of stable viral particles that efficiently infect target cells, but its effectiveness is hindered if specific requirements are not met. Conversely, viruses retained on producer cells can still undergo cell-to-cell transmission (Pais-Correia et al, 2010; Sherer et al, 2010). Furthermore, viruses that have been released from infected cells have the ability to be captured and stabilized by various components present on cell surfaces or within the extracellular matrix (Munch et al, 2007; Pais-Correia et al, 2010; Pan et al, 2007). Weak binding and insufficient activation of target cells impede cell-free spread (O'Doherty et al, 2000; Platt et al, 2010; van der Schaar et al, 2007), while abundant envelope glycoprotein at cell-to-cell interfaces enhances viral replication (Agosto et al, 2009; Balabanian et al, 2004; Vasiliver-Shamis et al, 2009; Yu et al, 2009). While viruses may not be optimally suited for cell-free propagation in certain conditions, it is important to acknowledge that cell-to-cell spread offers various advantages beyond being an alternative for less efficient viruses. Cell-to-cell spread is rapid, utilizing the cellular cytoskeleton for dissemination at contact sites. Enhanced budding at these sites promotes viral propagation even with low gene expression levels of proteins facilitating viral assembly (Jin et al, 2009). In addition, cell-to-cell spread allows viruses to evade neutralizing antibodies and overcome immunological and physical barriers within an organism (Ganesh et al, 2004; Hubner et al, 2009). Nevertheless, situations exist where cell-free virus propagation may confer greater advantages, as it is not constrained by specific cell-to-cell interactions and enables transmission between individuals. Hence, it is conceivable that specific viruses, like HIV, have developed mechanisms enabling them to switch between modes of propagation.

The challenge of comprehending virus spread through cell-to-cell transmission within living organisms highlights the crucial need to identify specific cellular factors and targeted inhibitors for each mode of transmission. Recent advancements have provided insights into the involvement of the cytoskeleton in cell-cell spreading mediated by contact, while the propagation of cell-free viruses follows distinct mechanisms (Fig. 3A).

### Cell-free virus particles

The release of viral particles into the extracellular milieu followed by their subsequent entry into new target cells represents a thoroughly comprehended mode of viral dissemination. This process is of paramount importance in facilitating viral spread among cells or between hosts. Virions discharged from an originating cell necessitate specific interactions with surface molecules on the recipient target cell. The release of cell-free viral particles is orchestrated through diverse mechanisms, including cell lysis induced by viral proteins, direct budding from the plasma

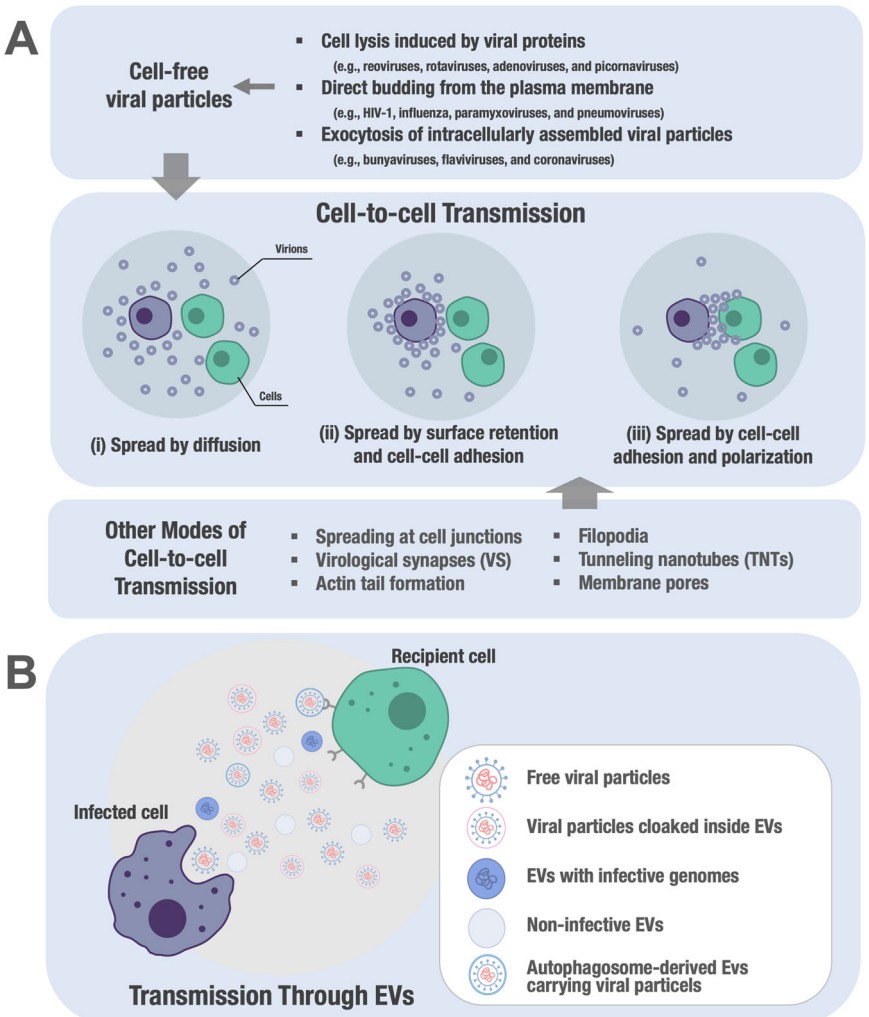

**Figure 3. Diverse viral transmission strategies.**

Viruses utilize cell-to-cell transmission strategies (**A**) or exploit extracellular vesicles (**B**) to facilitate the spread of the virus to target cells. (**A**) This figure summarizes how virus cell-to-cell transmission enhances viral spreading. (i) Virus diffusion creates a concentration gradient around infected cells, affecting the probability of viral particles reaching neighboring target cells. (ii) Retention of particles on donor cell surfaces increases local particle concentration, promoting infection of neighboring target cells. (iii) Optimal viral spread occurs when particles are polarized to cell-cell contact sites, facilitated by mechanisms involving cell-cell adhesion and polarity. This highlights the trade-off between long-range diffusion and efficient local viral spread. (**B**) This figure illustrates the interplay between extracellular vesicles (EVs), viruses, and various subpopulations of viral particles. Infected cells not only release replicative viral particles but also secrete structures containing viral proteins and nucleic acids, influencing immune responses and facilitating viral propagation. These particles, classified as host EVs containing viral molecules or defective viral particles, are challenging to isolate due to their similar size, density, and composition. Viral structures found within host EVs may include complete viral particles or "quasi-enveloped" viruses, where non-enveloped viruses are cloaked inside host EVs. Examples of enveloped viruses include HIV, influenza, dengue, and SARS-CoV-2, while non-enveloped viruses such as hepatitis A virus (HAV) and coxsackievirus typically promote cell lysis for release. The Trojan Horse hypothesis suggests that EVs secreted by infected cells can transmit viruses to recipient cells. Moreover, viruses exploit the exosomal pathway to enhance their transmission, as seen with various viruses including porcine reproductive and respiratory syndrome virus (PRRSV), herpes simplex virus (HSV-1), and enterovirus 71. This complex interaction underscores the role of EVs in viral spread and highlights potential immune evasion strategies employed by viruses.

membrane, or the exocytosis of intracellularly assembled viral particles (Cifuentes-Munoz et al, 2014; Giorda and Hebert, 2013; Weissenhorn et al, 2013).

### Spreading at cell junctions

Cell junctions, which comprise a diverse array of transmembrane proteins, serve as a means to establish a sealing barrier among polarized epithelial cells, thereby thwarting viral intrusion unless there is prior damage to the epithelial integrity (Mateo et al, 2015).

Viral particles have the capability to infect epithelial cells through either their apical or basolateral aspects, subsequently facilitating their transmission to neighboring cells located in proximity to partially sealed cell junctions. For instance, the hepatitis C virus (HCV) employs tight junction components such as CLDN1 (claudin 1) and OCLN (occludin), in conjunction with apolipoprotein E or receptors such as CD81, SR-BI (scavenger receptor class B, type I), and LDLR (low density lipoprotein receptor), to effectuate its spread (Brimacombe et al, 2011). In the context of

herpesviruses, adherens junctions represent the cellular structures exploited for mediating cell-to-cell transmission. Herpes simplex virus type 1 (HSV-1) utilizes nectin-1, a critical component of adherens junctions, as one of its receptors (Johnson and Baines, 2011). This process is guided by the glycoprotein complex known as gI/gE (glycoprotein I/glycoprotein E). Notably, The positioning of gI/gE complexes at the trans-Golgi network during early phases of infection aids in the organization of virions toward basolateral cell junctions. This process results in the coordinated release of viral particles, allowing them to enter neighboring cells. While the involvement of desmosomes in cell-to-cell spread is relatively less explored, its importance is highlighted by research concerning lymphocytic choriomeningitis virus (LCMV) (Labudova et al, 2019). In this context, keratin 1 acts as a stabilizing factor for desmosomes, strengthening cell-to-cell adhesion and thereby promoting the dissemination of LCMV.

### Virological synapses

Virological synapses (VS) represent a mechanism utilized for cell-to-cell transmission, particularly prominent among retroviruses such as human T-lymphotropic virus type 1 (HTLV-1) and HIV-1 (Dupont and Sattentau, 2020). Within the VS, a consortium of cellular components, encompassing cytoskeletal proteins and receptors, collaborates to expedite the swift conveyance of viral constituents from infected cells to uninfected counterparts (Igakura et al, 2003). This intricate process entails the recruitment of viral proteins and RNA to specific locales at cell-cell junctions, facilitated by the realignment of cellular structures like the microtubule organizing center (MTOC). In the case of HTLV-1, this reorientation is orchestrated by the viral Tax protein and necessitates the integrity of microtubules, actin filaments, and small GTPases (Nejmeddine et al, 2005). Analogously, cells infected with HIV-1 orchestrate the polarization of the MTOC and associated organelles towards the virological synapses. Within this specialized microenvironment, they engage the recruitment of CD4, CCR5, and CXCR4 receptors on the surface of uninfected cells (Bayliss et al, 2020). The conceptual model posits that viral particles undergo budding within the synaptic cleft, subsequently effecting fusion with neighboring uninfected cells. It is noteworthy that HIV-1 can concurrently disseminate to multiple adjacent cells, a phenomenon referred to as polysynapse formation. Robust in vivo investigations corroborate the pivotal role of VS in viral proliferation (Murooka et al, 2012; Rudnicka et al, 2009; Sewald et al, 2015). Intriguingly, the transmission mediated by VS appears to exhibit a reduced susceptibility to neutralizing antibodies, in contrast to cell-free transmission (Gombos et al, 2015; Zhong et al, 2013). Similar mechanisms of cell-to-cell transmission have been discerned in the context of other viruses such as murine leukemia virus (MLV) and non-lymphotropic herpes simplex virus type 1 (HSV-1), suggesting a wider applicability of VS-like modalities in viral propagation, a facet necessitating further scholarly exploration.

### Actin tail formation

Viral infections are disseminated through the transfer of viral particles from infected cells to uninfected target cells. Irrespective of the mode of transmission, a crucial element for viral spread is the reduction of viral receptor expression in infected cells (Neil et al, 2006; Pais-Correia et al, 2010; Sherer et al, 2010). This

downregulation, combined with a strong affinity for the receptor expressed on target cells, establishes a gradient that facilitates virus dissemination. When viruses encounter target cells, they strongly and specifically bind to their viral receptor, establishing a connection with the underlying actin cytoskeleton (Lehmann et al, 2005; Lidke et al, 2005). Rather than recruiting individual myosin motors, viruses exploit the general turnover of filamentous actin (F-actin) to move toward target cells (Burckhardt and Greber, 2009; Lidke et al, 2004; Medeiros et al, 2006; Schelhaas et al, 2008; Smith et al, 2008). This form of transmission based on affinity has been observed during interactions between MLV-infected fibroblasts and uninfected target cells (Sherer et al, 2010). Prolonged interactions result in the anchoring of target cell membranes, thereby facilitating virus assembly and utilizing retrograde F-actin flow to move toward target cells (Jin et al, 2009; Sherer et al, 2007). Viruses have evolved diverse strategies to exploit the actin cytoskeleton in target cells or infected cells, enabling efficient movement toward neighboring cells. A number of viruses employ the induction of actin tails to facilitate movement towards target cells, with some viruses forming actin tails after being released, while others generate them within the cytoplasm (Doceul et al, 2010; Jouvenet et al, 2006).

The concept of actin tail formation is deemed a shared strategy among orthopoxviruses, substantiated by sequence homology and corroborative evidence (Smith and Law, 2004). The replication of vaccinia virus (VACV) is orchestrated within cytosolic factories, yielding two distinct viral forms: intracellular mature virus (IMV) and extracellular enveloped virus (EEV). These viral forms exhibit structural, antigenic, and exit pathway disparities. IMVs are primarily liberated through host cell lysis, although budding becomes relevant during the late stages of infection. Conversely, EEV predominantly exits cells via plasma membrane budding and exocytosis. Within infected cells, actin-based structures referred to as "actin tails" emerge, initially localized intracellularly but eventually extending up to 20 µm from the cellular surface (Welch and Way, 2013). EEVs positioned at the termini of these actin tails facilitate direct cell-to-cell transmission (Donnelly et al, 2013). VACV proteins A36 and A33 can induce the formation of actin tails, augmenting virus dissemination to adjacent uninfected cells. Cellular factors, including clathrin, AP-2, and CK2, amplify the formation of actin tails (Duncan et al, 2018; Welch and Way, 2013). Actin tails comprise two cytoplasmic actin isoforms, β and γ-actin, with β-actin being the principal contributor to actin nucleation prompted by VACV (Marzook et al, 2017).

In addition, a proposed mode of viral transmission through nanotubes exists, although further verification is required (Eugenin et al, 2009; Gerdes and Carvalho, 2008; Sherer and Mothes, 2008). It is suggested that viruses like HIV may travel along the outer surface of nanotubes, similar to how MLV moves along filopodial bridges (Sowinski et al, 2008).

### Tunneling nanotubes

Tunneling nanotubes (TNTs) represent slender, elongated membrane-bound conduits, ranging from 50 to 200 nm in width. These structures serve as bridges between remote cells, facilitating the transfer of diverse materials, including proteins, organelles, miRNAs, and ions. A comprehensive review on TNTs has recently been published (Jansens et al, 2020). Within macrophages, two distinct TNT categories are discerned based on their composition:

"thin TNTs", housing F-actin, and "thick TNTs" (exceeding 0.7 μm), incorporating both F-actin and microtubules. Unlike their thin counterparts, thick TNTs accommodate organelles and even facilitate the movement of bacteria along their surfaces (Onfelt et al, 2006). Viruses effectively harness TNTs for cell-to-cell transmission, as vividly demonstrated by HIV (Sowinski et al, 2008). These F-actin-rich TNTs establish connections between distant T cells and facilitate rapid, receptor-dependent HIV propagation. This phenomenon of TNT induction by HIV is similarly observed in macrophages, effectively distinguishing these structures from filopodia (Eugenin et al, 2009). HTLV-1 exhibits the capacity to create conduits in T cells closely resembling TNTs, a process contingent upon viral proteins and cellular factors (Van Prooyen et al, 2010). Various other viruses, including influenza and PRRSV, exploit TNTs to achieve intercellular transmission. Influenza genetic material and proteins traverse TNTs without impediment from antibodies or antiviral agents, signifying their open-ended nature (Kumar et al, 2017). Moreover, pseudorabies virus prompts the formation of exceptionally stable TNTs that facilitate viral spread, enabling the transportation of virions within these structures, subsequently released through exocytosis at the contact sites with recipient cells (Jansens et al, 2017). Lastly, DNA viruses like VACV and bovine herpesvirus 1 (BoHV-1) elicit TNT-like structures. These structures feature the presence of viral proteins along their length, enabling the direct transfer of viral material (Panasiuk et al, 2018).

### Filopodia

Filopodia are cellular structures with diverse functions, encompassing roles in processes like cell migration, wound healing, extracellular matrix adhesion, and cell-cell interactions (Blake and Gallop, 2023). They arise from actin polymerization beneath the plasma membrane, resulting in slender extensions typically measuring 0.1–0.3 μm in diameter, characterized by parallel actin bundles. Filopodia formation is initiated by Rho GTPases, notably Cdc42 (Faix and Rottner, 2006). Identifying filopodia from tunneling nanotubes (TNTs) can be intricate due to their heterogeneous nature, and it has been suggested that some TNTs may originate from filopodium-like protrusions connecting adjacent cells. Recent findings suggest the possibility of two filopodial connections interacting through N-cadherin, resulting in the formation of a bridge-like structure (Rustom et al, 2004). Actin filament rotation within these bridges, driven by myosin proteins, subsequently leads to the separation of filopodia. Subsequently, one of these filopodia elongates toward a cell body (Delage et al, 2016), establishing a stable TNT-like connection facilitated by N-cadherin/β-catenin clustering. Murine leukemia virus (MLV) exploits filopodia during its infection cycle through a phenomenon referred to as "virus surfing" (Lehmann et al, 2005). MLV particles initially bind to cellular receptors and navigate along filopodial extensions to access the host cell's body, thereby facilitating viral entry. In later stages of infection, viral particles traverse directly between infected and uninfected cells employing filopodial bridges known as viral cytonemes, which typically measure approximately 5.8 μm in length. The formation of these cytonemes hinges on interactions between MLV's Env protein and host cell receptors, and their integrity can be compromised by antibodies targeting Env's extracellular domain. Other viruses, including herpesviruses and alphaviruses, induce filopodia-like projections characterized by

distinct features, some of which transport viral particles (Dixit et al, 2008; Smith et al, 2008). These projections can enhance viral spread by evading neutralizing antibodies. Respiratory viruses such as respiratory syncytial virus (RSV) and human metapneumovirus (HMPV) also trigger the formation of filopodial structures, which augment viral dissemination (El Najjar et al, 2016). Within cells infected by HMPV, the branching filaments could potentially represent filamentous virus structures. RSV-induced filopodia formation is dependent on both the cellular actin-related protein 2 (Arp2) and the RSV F protein (Mehedi et al, 2016). In addition, severe acute respiratory syndrome virus 2 (SARS-CoV-2) has been associated with the induction of filopodia harboring budding viral particles, potentially mediated by casein kinase 2 signaling (Bouhaddou et al, 2020).

### Membrane pores

The exploration of measles virus (MeV) infection in primary human airway epithelial (HAE) cells has brought to light an innovative pathway for direct cell-to-cell transmission characterized by the presence of membrane pores. In contrast to certain other viruses, MeV avoids syncytia formation but efficiently disseminates within HAE cell cultures, culminating in the formation of larger infectious centers than other viruses (Singh et al, 2015). This propagation mechanism hinges on the presence of intercellular pores, approximately 250 nm in diameter. The formation of these infectious centers relies on the interaction between nectin-4, a receptor responsible for governing MeV entry, and afadin, an actin filament binding protein (Singh et al, 2016). Recent findings revealed an alternative method of spreading utilized by MeV (Generous et al, 2019). This method entails cells expressing nectin-1 capturing membranes containing nectin-4, a process termed "trans-endocytosis". Through this mechanism, various cytosolic components, including MeV ribonucleocapsids, are transferred between cells. This transmission mode operates effectively, even when involving primary neurons, and does not necessitate the formation of complete virions.

## Intercellular transmission through extracellular vesicles

Extracellular vesicles (EVs) have recently emerged as a novel mechanism utilized by non-enveloped viruses to exit host cells without causing cell lysis, thereby facilitating viral transmission (Fig. 3B). By exploiting extracellular vesicles, multiple viral particles can collectively move into and out of cells, thereby enhancing viral adaptability, expanding transmission pathways, and evading immune detection. EVs are membrane-bound particles composed of lipid bilayers that are naturally released from cells (van Niel et al, 2018). They act as carriers, transporting a diverse range of cargo molecules such as nucleic acids, proteins, metabolites, lipids and even virus particles to recipient cells. These vesicles undergo diverse formation processes. Firstly, they can emerge directly from the plasma membrane, resulting in the creation of microvesicles or ectosomes. Alternatively, during the formation of multivesicular bodies (MVBs), intraluminal vesicles (ILVs) bud and subsequently release as exosomes into the extracellular environment through fusion between MVBs and the plasma membrane. Another mechanism involves the release facilitated by autophagosomes, where single-membrane vesicles are released by the fusion of double-membrane autophagosomes with the plasma membrane,

without causing cell lysis. Lastly, apoptotic processes trigger the generation of apoptotic bodies.

Traditionally, it was believed that non-enveloped viruses were released from infected cells through cell lysis. However, in the past decade, a multitude of studies have revealed that various naked RNA viruses employ EVs as a non-lytic pathway to exit host cells. This phenomenon has been observed in several members of the *Enterovirus* genus and the *Picornaviridae* family, which includes viruses like Enterovirus 71 (Gu et al, 2020; Mao et al, 2016), Rhinovirus (Chen et al, 2015), Coxsackievirus B3 (Robinson et al, 2014), and Poliovirus (Bird et al, 2014). In addition, viruses from the *Cardiovirus* genus, such as encephalomyocarditis virus, and the *Hepatovirus* genus, represented by hepatitis A virus (HAV), as well as viruses belonging to other families like rotavirus (*Reoviridae*), norovirus (*Caliciviridae*), and hepatitis E virus (*Hepeviridae*) employ EVs as part of their exit strategy from host cells (Feng et al, 2013; Nagashima et al, 2014; Santiana et al, 2018; van der Grein et al, 2019). These extracellular vesicles either encompass plasma membranes or autophagosomal membranes and preserve infectivity upon transfer to recipient cells. Further supporting the role of extracellular vesicles in viral spread, evidence has emerged regarding the non-lytic release of poliovirus capsids (Bird et al, 2014). Large extracellular vesicles, enriched with phosphatidylserine (PS) and measuring up to 500 nm, house multiple mature poliovirus particles. A similar phenomenon is observed in other enteroviruses such as rhinovirus and coxsackievirus B3 (CVB3) (Chen et al, 2015). These poliovirus-containing vesicles enter target cells dependent on both the presence of PS and the cellular receptor CD155 on the vesicle membranes. Poliovirus particles enclosed within vesicles display enhanced infectivity per particle in comparison to cell-free viral particles, possibly attributed to the combined effect of multiple particles entering cells concurrently. Non-lytic discharge of vesicles containing rotavirus has been observed in cultured cells. The significance of this process is underscored by the detection of vesicles carrying clusters of rotavirus (typically 15 active viruses) in animal fecal samples (Santiana et al, 2018). Similar to poliovirus-containing vesicles, rotavirus enclosed within vesicles displays heightened infectivity compared to free viral particles. In addition, it has been confirmed that norovirus particles can be liberated non-lytically within vesicles, albeit these vesicles are slightly smaller in size and exhibit cellular markers resembling exosomes. The first documented cases of viral particles enclosed within exosomal membranes were noted in research involving HCV and HAV (Ramakrishnaiah et al, 2013). These exosomes, which carry viral particles, were proved to transmit HCV infection to recipient cells, and this process displayed some resistance to neutralizing antibodies. Similarly, vesicles with exosome markers were observed to house multiple HAV particles, facilitating their transfer between liver cells. Quasi-enveloped HAV (eHAV) particles, which are partially shielded by membranes, exhibit resistance to neutralizing antibodies and have the ability to enter target cells by a process dependent on dynamin and clathrin (Rivera-Serrano et al, 2019). Upon endocytosis, eHAV particles are identified by receptor gangliosides in endolysosomes, initiating their transport to lysosomes where the exosomal membrane is dismantled, enabling viral uncoating to occur (Das et al, 2020). These novel mechanisms of spread offer viruses an advantage because neutralizing antibodies are less potent against

**Box 1  In need of answers**

- What are the specific mechanisms by which neurotropic viruses induce behavioral alterations in infected hosts?

- How do these behavioral changes influence the transmission dynamics of various viral pathogens?

- Are there differences in the transmission efficacy of viruses depending on the nature and extent of behavioral modifications induced in their hosts?

- What are the implications of these findings for the development of targeted interventions to control viral spread?

vesicles than against cell-free particles, mainly due to the protective vesicular membrane. This phenomenon is illustrated by the human JC polyomavirus, which also interacts with EVs. In addition, it has been demonstrated that vesicles containing the JC polyomavirus can enter target cells without relying on cellular receptors. Therefore, the generation of extracellular vesicles housing infectious virus particles seems to be a common mechanism employed by many RNA and DNA viruses, irrespective of their enveloped or non-enveloped nature (Altan-Bonnet et al, 2019).

The utilization of EVs by naked viruses offers several advantageous features (Altan-Bonnet, 2016; Altan-Bonnet et al, 2019; Feng and Lemon, 2014). These include the capacity to exit infected cells through non-lytic pathways, thereby enabling virus release without causing cell lysis. By exploiting EVs, viruses can diversify their transmission routes, leading to heightened dissemination of the viral particles. Moreover, EV-mediated transport facilitates the coordinated delivery of viral components, promoting genetic cooperativity and ultimately enhancing viral fitness. Furthermore, the inclusion of non-enveloped viruses within EVs functions as a protective mechanism against neutralizing antibodies that specifically recognize and target the viral capsid.

## Summary and perspectives

Viruses have evolved a remarkable repertoire of mechanisms to manipulate and exploit their hosts and vectors, enabling their efficient transmission and long-term survival. By unraveling the intricacies of these mechanisms, we can gain valuable insights into the captivating biology of viruses. Such insights have the potential to guide the development of innovative and targeted approaches for the treatment and prevention of viral infections, thereby safeguarding public health. In addition, as highlighted in Box 1, open questions remain unanswered regarding the specific mechanisms by which viruses induce behavioral alterations in infected hosts or vectors, how these alterations influence transmission dynamics, and the implications of such findings for the development of interventions.

## Peer review information

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

## Acknowledgements

This work was funded by grants from the National Key Research and Development Plan of China (2021YFC2300200, 2022YFC2303200, 2021YFC2302405 and 2022YFC2303400), the National Natural Science Foundation of China (32188101 and 82102389), the Shenzhen San-Ming Project for Prevention and Research on Vector-borne Diseases (SZSM202211023), and the Science and Technology Project of Southwest United Graduate School of Yunnan (202302AO370010). The New Cornerstone Science Foundation through the New Cornerstone Investigator Program, and the Xplorer Prize from Tencent Foundation.

## Author contributions

**Xi Yu**: Writing—original draft; Writing—review and editing. **Yibin Zhu**: Supervision; Writing—review and editing. **Gang Yin**: Resources. **Yibaina Wang**: Resources. **Xiaolu Shi**: Resources. **Gong Cheng**: Supervision; Project administration; Writing—review and editing.

## Disclosure and competing interests statement

The authors declare no competing interests.

