## [Peer Review File · EMBO Reports]

Exploiting Hosts and Vectors: Viral Strategies for Facilitating Transmission

Xi Yu, Yibin Zhu, Gang Yin, Yibaina Wang, Xiaolu Shi, and Gong Cheng

Corresponding author(s): Gong Cheng (gongcheng@mail.tsinghua.edu.cn)

Review Timeline:

Submission Date:	29th May 23
Editorial Decision:	24th Aug 23
Revision Received:	9th Oct 23
Editorial Decision:	15th Mar 24
Revision Received:	17th Apr 24
Accepted:	25th Jun 24

Editor: *Martina Rembold*

Transaction Report:

Dear Prof. Cheng,

Thank you for the submission of your review to EMBO reports. I apologize for the delay in handling your manuscript but it took longer than usual to secure enough referees. We have now received the two enclosed reports on it.

As you will see, referee 2 provides detailed and very constructive feedback. Reading the report, it becomes clear that a significant revision will be required before the review article can be considered for publication in EMBO Reports. As it stands, the focus of the review is too broad to appeal to our readership and the coverage of mainly older studies limits the timeliness of your review article. Moreover, the article does not provide an in-depth discussion of the various topics and is often limited to the description of one virus type.

I still think that in particular the first section on how viruses alter the behavior of their hosts or vectors is interesting and timely and I would therefore like to offer you to revise your review along the lines suggested by referee 2 and summarized again below:

- Please shorten and focus the review on how viruses change the behavior of the host and on unconventional viral transmission strategies. The second part should be removed.
- Provide more in-depth information on the molecular mechanisms that drive the change in host and vector behavior.
- Expand the third part on cell-to-cell spread to other virus systems and add more information and in-depth discussion of virus transmission by extracellular vesicles.
- I also fully agree with the statement from referee 2 that less than 5 percent of cited papers are from the last 5 years, which I also noted when I first read your review. For a timely review, more recent studies must be included.
- Also, as the referee pointed out, more figures providing schematic summaries for all topics covered in the review are needed. You can include up to 5 figures and it will greatly strengthen the value of your review if you graphically depict major concepts and/or mechanisms.

I think the review has the potential to constitute an interesting contribution to the field and while I appreciate that incorporating the referees' suggestions will require a bit of work, I am convinced that the article is worth it and will benefit from it.

Please submit along with your review a detailed point-by-point response to the referee suggestions to facilitate the evaluation of your review.

When submitting your revised manuscript, we will require a Microsoft Word file (.docx) of the revised manuscript text including detailed figure legends, but without the figures. Finally, please also provide an extra file with the figure sketches. Please also add a statement specifying whether or not authors have competing interests (defined as all potential or actual interests that could be perceived to influence the presentation or interpretation of an article). In case of competing interests, this must be specified in your disclosure statement. Further information: <https://www.embopress.org/competing-interests>

Thank you again for writing this piece for us, and please let me know if you have any questions regarding the revisions.

I look forward to seeing a revised version of your manuscript when it is ready. Please let me know if you have questions or comments regarding the revision.

Kind regards,

Referee #1:

In this manuscript, Xi Yu and his/her colleagues systematically illustrated the transmission strategies of viruses. They gave sufficient details from three main aspects: i) virus-mediated manipulation of host and vector behaviors; ii) Immunol or antiviral evasion by viruses; iii) cell-to-cell and extracellular vesicle-mediated transmission. There are detailed and convincing demonstration for each part. I do not have any concerns or questions and think it is suitable for publication.

Referee #2:

In this manuscript, Yu and colleagues review various aspects of how viruses manipulate their host to facilitate virus spread. The authors discuss three aspects in three subsequent sections.

The first section covers the topic how viruses can change the behaviour of the host or the transmitting arthropod vector. This section of the review covers a topic that so far has not been widely reviewed and therefore, I consider this section as novel and informative. That said, I find this section difficult to read. It would have been much easier to follow if the authors would have provided a somewhat detailed schematic summary of this particular section. Moreover, I was missing a bit more in-depth discussion of molecular mechanisms driving these behavioural changes.

In the next section of the review, the authors discuss "Viral Immune Evasion: Manipulating Cellular Machinery for Survival". This section focusses on herpesviruses and discusses textbook knowledge. Unfortunately, the authors do not go beyond the textbook and mainly discuss literature from the 1990s. I find this section of the review rather outdated and more importantly, there are numerous more up-to-date reviews to this particular topic. Therefore, I suggest to remove this whole chapter from the review. Instead, the authors should discuss in more depth the first and third section.

The third section of the review discusses unconventional viral transmission strategies. In the first part, the authors review cell-to-cell spread. While this is an interesting and not so heavily covered topic, the authors limit their view very much to HIV and vaccinia virus. While these are well-studied examples, it would be helpful to expand this section to other virus systems for which cell-to-cell spread has been intensively studied (e.g. hepatitis C virus). The subsequent section on virus transmission by extracellular vesicles is a most recently emerging topic. Unfortunately, the authors do not discuss this in sufficient depth. It is stated that certain viruses are released via EVs, yet no description of mechanisms and consequences is provided. Moreover, the authors do not discuss that viral genomes can be released via EVs providing yet another means of (virus particle independent) spread. This needs to be included.

Overall, while this review aims to provide a broad understanding of viral transmission mechanisms, it would have benefited from a more focused and in-depth discussion, especially related to the first and third section of the review that cover much less intensively reviewed topics. The authors' attempt to cover a wide range of topics limited their ability to delve more deeply into molecular mechanisms, which I find a major limitation of this review.

Further points:

- Overall, less than 5 percent of cited papers are from the last 5 years, which I find a bit surprising. For a timely review, more recent papers should be included. Therefore, the authors should revisit their references and ensure that all statements are based on the latest studies.
- Following the introduction, the authors present a visually appealing figure that has the character of a graphical summary as it summarizes the various aspects of the review in a rather superficial, yet graphically appealing manner. However, this figure contains several aspects that cannot be fully understood by the reader in the introduction since an explanation is provided only later in the article. Therefore, this figure would fit better at the end of the review as kind of a summary (yet, requires a much more detailed legend).
- The review would be much better digestible, if the authors would include a more detailed schematic for each of the main sections. For instance, the text related to the behavioural changes of the vector as a result of viral infection (of the host or the vector itself), is lengthy and would be much more appealing if the authors would include a somewhat detailed graphical summary illustrating the various aspects that are discussed in this section.
- Page and line numbers would have been helpful.
- Page 2: "In its simplest manifestation, virus-mediated lysis of host cells enables the unrestricted movement of viruses through the surrounding medium or extracellular matrix via Brownian motion, ultimately reaching new host cells." This also applies to viruses that are released by a non-cytolytic mechanism.
- Page 3: Here the authors often write "viruses" but this section is mainly focussed on rabies virus. The authors should include other examples.
- Page 4: "Although a detailed structural model of the rabies virus glycoprotein is currently unavailable, the ..." That is not true; see e.g. DOI: 10.1126/sciadv.abp9151
- Page 8: "The vertebrate host provides an advantageous milieu for viruses, facilitating their replication, ..." Why only the vertebrate host?
- Page 9: "MHC class I molecules survey the cytosol and present ..." Sounds strange, especially for an ER-luminal protein.
- Page 12: "Enveloped viruses are characterized by the presence of an extra lipid bilayer that surrounds" Why extra? Most enveloped viruses have just one lipid bilayer.
- Page 12: "Within host cells, they undergo genetic replication, assembly..." I suggest to delete "genetic" or do the authors mean

genome replication?

- Page 13: What do the authors mean with "Enhanced budding at these sites facilitates propagation even with low gene expression levels (Jin et al, 2009)." It is not clear to me what the authors want to state. Gene expression of the host cell?

Response to Editor's Comments:

Point 1: Please shorten and focus the review on how viruses change the behavior of the host and on unconventional viral transmission strategies. The second part should be removed.

Response 1: We have shortened and restructured the review to primarily emphasize how viruses alter host/vector behavior and unconventional viral transmission strategies. Specifically, we have expanded the discussion in the first section (Page 3, Lines 95-106; Page 4, Lines 118-155) and the third section (Pages 10-14, Lines 418-571; Pages 15-16, Lines 600-638). The second section, as per your request, has been removed to maintain a more concise focus (Page 8, Line 332).

Point 2: Provide more in-depth information on the molecular mechanisms that drive the change in host and vector behavior.

Response 2: To enhance the depth of information on the molecular mechanisms, we have expanded the relevant sections. By delving into in-depth explanations of the molecular mechanisms involved and further engaging in discussions about various viruses, we are confident that we are enhancing the comprehensiveness of our coverage of these processes. (Page 3, Lines 95-106; Page 4, Lines 118-155).

Point 3: Expand the third part on cell-to-cell spread to other virus systems and add more information and in-depth discussion of virus transmission by extracellular vesicles.

Response 3: We have broadened the discussion and added a new section discussing different modes of cell-to-cell spread across various virus systems and incorporated an in-depth examination of virus transmission via extracellular vesicles (Pages 10-14, Lines 418-571; Pages 15-16, Lines 600-638). This addition should provide readers with a more thorough insight into these transmission modes.

Point 4: I also fully agree with the statement from referee 2 that less than 5 percent of cited papers are from the last 5 years, which I also noted when I first read your review. For a timely review, more recent studies must be included.

Response 4: We have diligently incorporated more recent studies to ensure that the review remains up-to-date and reflective of the current state of research in the field.

Point 5: Also, as the referee pointed out, more figures providing schematic summaries for all topics covered in the review are needed. You can include up to 5 figures and it will greatly strengthen the value of your review if you graphically depict major concepts and/or mechanisms.

Response 5: We have included three more figures that visually summarize key concepts and mechanisms discussed throughout the review (Figure2-4). These figures should enhance the overall clarity and value of the article.

Response to Reviewer 1 Comments:

Comment: In this manuscript, Xi Yu and his/her colleagues systematically illustrated the transmission strategies of viruses. They gave sufficient details from three main aspects: i) virus-mediated manipulation of host and vector behaviors; ii) Immunol or antiviral evasion by viruses; iii) cell-to-cell and extracellular vesicle-mediated transmission. There are detailed and convincing demonstration for each part. I do not have any concerns or questions and think it is suitable for publication.

Response: Thank you for your thorough review of our manuscript and we appreciate your positive feedback. We have taken the other reviewer's comments into account and have made several revisions to ensure the clarity and accuracy of our manuscript. We have shortened and restructured the review to primarily emphasize how viruses alter host/vector behavior and unconventional viral transmission strategies (Page 8, Line 332). Furthermore, we have broadened the discussion and added a new section discussing different modes of cell-to-cell spread across various virus systems and incorporated an in-depth examination of virus transmission via extracellular vesicles (Page 3, Lines 95-106; Page 4, Lines 118-155; Pages 10-14, Lines 418-571; Pages 15-16, Lines 600-638). We have also included more figures that visually summarize key concepts and mechanisms discussed throughout the review (Figure2-4).

Response to Reviewer 2 Comments:

Point 1: This section of the review covers a topic that so far has not been widely reviewed and therefore, I consider this section as novel and informative. That said, I find this section difficult to read. It would have been much easier to follow if the authors would have provided a somewhat detailed schematic summary of this particular section. Moreover, I was missing a bit more in-depth discussion of molecular mechanisms driving these behavioural changes.

Response 1: We would like to express our gratitude for your thoughtful review and valuable feedback. We acknowledge your suggestion regarding the inclusion of a schematic summary for this section. We have incorporated a more detailed visual representation to enhance the clarity and facilitate a better understanding of the concepts discussed (Figure 2 and 3). This visual aid will provide readers with a visual roadmap, making it easier to follow the content. To enhance the depth of information on the molecular mechanisms, we have expanded the relevant sections. By delving into in-depth explanations of the molecular mechanisms involved and further engaging in discussions about various viruses, we are confident that we

are enhancing the comprehensiveness of our coverage of these processes. (Page 3, Lines 95-106; Page 4, Lines 118-155).

Point 2: I find this section of the review rather outdated and more importantly, there are numerous more up-to-date reviews to this particular topic. Therefore, I suggest to remove this whole chapter from the review. Instead, the authors should discuss in more depth the first and third section.

Response 2: We have shortened and restructured the review to primarily emphasize how viruses alter host/vector behavior and unconventional viral transmission strategies. Specifically, we have expanded the discussion in the first section (Page 3, Lines 95-106; Page 4, Lines 118-155) and the third section (Pages 10-14, Lines 418-571; Pages 15-16, Lines 600-638). The second section, as per your request, has been removed to maintain a more concise focus (Page 8, Line 332).

Point 3: The third section of the review discusses unconventional viral transmission strategies. In the first part, the authors review cell-to-cell spread. While this is an interesting and not so heavily covered topic, the authors limit their view very much to HIV and vaccinia virus. While these are well-studied examples, it would be helpful to expand this section to other virus systems for which cell-to-cell spread has been intensively studied (e.g. hepatitis C virus). The subsequent section on virus transmission by extracellular vesicles is a most recently emerging topic. Unfortunately, the authors do not discuss this in sufficient depth. It is stated that certain viruses are released via EVs, yet no description of mechanisms and consequences is provided.

Response 3: We acknowledge the importance of including a broader range of virus systems that exhibit cell-to-cell spread. In our revised manuscript, we have broadened the discussion and added a new section discussing different modes of cell-to-cell spread across various virus systems (Pages 10-14, Lines 418-571). We appreciate your point about the need for a more in-depth discussion of virus transmission via extracellular vesicles (EVs). In our revised version, we have delved deeper into the mechanisms involved in this process and discuss the consequences of such transmission (Pages 15-16, Lines 600-638). This will allow readers to gain a better understanding of this emerging topic and its significance.

Point 4: Overall, less than 5 percent of cited papers are from the last 5 years, which I find a bit surprising. For a timely review, more recent papers should be included. Therefore, the authors should revisit their references and ensure that all statements are based on the latest studies.

Response 4: We appreciate your feedback regarding the currency of the references cited in our review. We have thoroughly revisited our reference list and prioritized the inclusion of more recent studies. We also removed a couple of outdated references. We recognize the importance of grounding our statements in the latest research findings to provide readers with the most up-to-date and accurate information.

Point 5: Following the introduction, the authors present a visually appealing figure that has the character of a graphical summary as it summarizes the various aspects of the review in a rather superficial, yet graphically appealing manner. However, this figure contains several aspects that cannot be fully understood by the reader in the introduction since an explanation is provided only later in the article. Therefore, this figure would fit better at the end of the review as kind of a summary (yet, requires a much more detailed legend). The review would be much better digestible, if the authors would include a more detailed schematic for each of the main sections. For instance, the text related to the behavioural changes of the vector as a result of viral infection (of the host or the vector itself), is lengthy and would be much more appealing if the authors would include a somewhat detailed graphical summary illustrating the various aspects that are discussed in this section.

Response 5: We are grateful for the reviewer's feedback and have taken it into account. In response, we have included more figures that visually summarize key concepts and mechanisms discussed throughout the review (Figure2-4). These figures should enhance the overall clarity and value of the article.

Point 6: Page and line numbers would have been helpful.

Response 6: We have included page and line numbers in the revised manuscript.

Point 7: Page 2: "In its simplest manifestation, virus-mediated lysis of host cells enables the unrestricted movement of viruses through the surrounding medium or extracellular matrix via Brownian motion, ultimately reaching new host cells." This also applies to viruses that are released by a non-cytolytic mechanism.

Response 7: Thank you for your valuable comment on our manuscript. We acknowledge that this concept also applies to viruses released by non-cytolytic mechanisms. In light of your feedback, we have revised the expression as "In its simplest manifestation, viruses benefit from the unrestricted movement through the surrounding medium or extracellular matrix via Brownian motion, ultimately reaching new host cells. (Page 2, Lines 41-43)"

Point 8: Page 4: "Although a detailed structural model of the rabies virus glycoprotein is currently unavailable, the ..." That is not true; see e.g. DOI: 10.1126/sciadv.abp9151

Response 8: Thank you for bringing this to our attention. We have removed this expression accordingly (Page 5, Line 121).

Point 9: Page 12: "Enveloped viruses are characterized by the presence of an extra lipid bilayer that surrounds" Why extra? Most enveloped viruses have just one lipid bilayer.

Response 9: We apologize for the confusion in our expression and have removed the word "extra" (Page 10, Line 342).

Point 10: Page 12: "Within host cells, they undergo genetic replication, assembly..." I suggest to delete "genetic" or do the authors mean genome replication?

Response 10: Thank you for the suggestion and we have removed the word "genetic" (Page 10, Line 343).

Point 11: Page 13: What do the authors mean with "Enhanced budding at these sites facilitates propagation even with low gene expression levels (Jin et al, 2009)." It is not clear to me what the authors want to state. Gene expression of the host cell?

Response 11: In the cited study of murine leukemia virus, the authors revealed that "Gag proteins were drawn to adhesive zones formed by viral Env glycoprotein and its cognate receptor to promote virus assembly at cell-cell contact." To avoid confusion, we have modified this sentence as "Enhanced budding at these sites promotes viral propagation even with low gene expression levels of proteins facilitating viral assembly. (Pages 11-12, Lines 388-390)"

Dear Prof. Cheng,

Thank you for the submission of your revised review to EMBO reports. I apologize for this unusual delay in handling your manuscript! We have now received the comments from referee #2 that are pasted below.

As you will see, the referee appreciates that the revision has significantly improved the article but notes some remaining concerns and suggestions how to strengthen the first chapter. Please either provide specific evidence how behavior changes increase virus transmission in human/mammalian hosts or modify and change the title of this chapter and acknowledge and discuss that direct evidence for an effect on transmission in the context of e.g., rabies, is missing. In this context I also note that we need a short paragraph called "Box 1: In need of answers", that allows you to describe open questions in the field in the form of a bullet pointed list.

As standard procedure, I have gone through the text and introduced some changes and suggestions that I kindly ask you to consider (modified text attached).

Please also address the following editorial points, before we can proceed with the formal acceptance.

- Figure 1 looks really great, but it is also repetitive with Figure 2 - 4. I like the virus using the host as puppet. Maybe panel A - C can be modified and made 'simpler' to provide just some sort of idea of behavior changes, changes to host attractiveness, and general mechanisms of viral spread.

- Please add up to five keywords.

- Regarding the Author Contributions, we now use CRedit to specify the contributions of each author in the journal submission system. Therefore, please remove the Author Contributions from the manuscript file and make sure that the author contributions in our manuscript tracking system are correct and up-to-date. The information you specified in the system will be automatically retrieved and typeset into the article. You can enter additional information in the free text box provided, if you wish.

- Please specify the following funding sources in our online manuscript tracking system in addition to the text: Tsinghua University Spring Breeze Fund (2020Z99CFG017), Shenzhen Science and Technology Project (JSGG20191129144225464), the Yunnan Cheng gong expert workstation (202005AF150034), the Innovation Team Project of Yunnan Science and Technology Department (202105AE160020), Tsinghua-Foshan Innovation Special Fund(TFISF) (2022THFS6124) and Shenzhen San-Ming Project for Prevention and Research on Vector-borne Diseases, XPLOER PRIZE from Tencent Foundation.

Thank you again for writing this piece for us, and please let me know if you have any questions regarding the revisions.

I look forward to seeing a revised version of your manuscript when it is ready.

Yours sincerely,

Referee #2:

The revised review by Yu and colleagues is a clear improvement over the previous version and omits the detailed description of immunological aspects that are already heavily covered by various reviews. Apart from a few more minor points, I still have one major concern, which relates to the first main chapter, i.e. "Viral Manipulation of Host and Vector Behaviors: Strategies for Transmission Maximization". Here, the authors dedicate an extensive chapter on host behaviour changes induced by viral infections, which is by and large a description of the pathogenesis of neurotropic viruses. The authors summarize mainly findings from rabies and herpes virus studies, how these viruses infect the CNS and that in some cases changes in the behaviour of the infected organism are resulting from that. However, it remains completely unclear if and to what extent these neurological changes impact virus transmission (the headline indicates "strategies for transmission maximization"). The only hint I found is "This aggressive conduct can potentially facilitate the transmission of the infection to new hosts", but I wonder whether the authors can be more specific on this aspect, which is the main topic of this review. In the case of herpesviruses, it is obvious that latency and reactivation facilitate transmission, but this is not linked to changes in behaviour, but rather due to the persistence that allows prolonged and repeated virus shedding. Therefore, I would encourage the authors to elaborate on the

(reported) impact of increased transmission (probably best studied in the context of rabies virus) or make clear that we do not know the real role behaviour changes might have for increased virus transmission. In the current version, this paragraph briefly summarizes the CNS pathogenesis of rabies and herpes viruses, but leaves open which role it might play regarding increased virus transmission.

Further points:

- L.32: Viruses are obligate intracellular parasites...
- L.70: Please rephrase because it reads as if viruses are prokaryotic pathogens.
- L.76 onward: In this chapter, the term "parasitic" is used somewhat ambiguous, because it reads like parasite, but what is meant are viruses. For instance, L.77 indicates parasite-induced alterations in host behaviour, but it's all about virus-induced changes. Therefore, better avoid the term parasitic in all these contexts.
- L.108: The studies demonstrate ...
- L.337: Not all enveloped viruses have a capsid. Some have only an RNP or even lack a capsid (e.g. GBV).
- L.437: apolipoprotein E is not a receptor
- L.578 onward: The authors may want to state that "They act as carriers, transporting a diverse range of cargo molecules such as nucleic acids, proteins, metabolites, lipids and even virus particles."

All editorial and formatting issues were resolved by the authors.

Prof. Gong Cheng
Tsinghua University
School of Medicine
Rm 4301, Biotech Building
Beijing 100084
China

Dear Gong,

Thank you for agreeing to the final changes to the review figures. I am pleased to inform you that your manuscript has been accepted for publication in EMBO reports. Thank you very much for contributing this review article to EMBO reports. I was a pleasure to work with you on it.

Your manuscript will be processed for publication by EMBO Press. It will be copy edited and you will receive page proofs prior to publication. Please note that you will be contacted by Springer Nature Author Services to complete licensing information.

When you receive an email from Springer Nature asking you to sign your license agreement, please enter the following code on the payment screen which should remove any charges due: LTE0ODI1NJUWMJM

If you have any questions, please do not hesitate to contact the Editorial Office. Thank you very much for your contribution to EMBO Reports.

Kind regards,

Martina
